# Think Silently, Think Fast: Dynamic Latent Compression of LLM Reasoning Chains

**Wenhui Tan**[1]*    **Jiaze Li**[2]    **Jianzhong Ju**[2]    **Zhenbo Luo**[2]    **Ruihua Song**[1]✉

**Jian Luan**[2]✉

[1]Gaoling School of Artificial Intelligence, Renmin University of China, Beijing, China
[2]MiLM Plus, Xiaomi Inc., Beijing, China

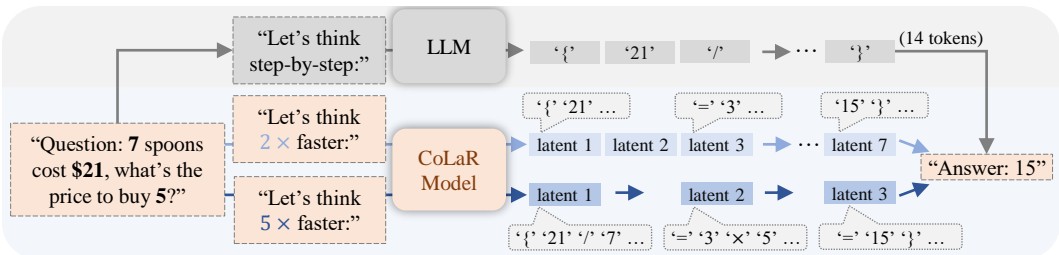

Figure 1: Our proposed Compressed Latent Reasoning Model (CoLaR) performs dynamic-speed reasoning by auto-regressively predicting latent variables, each compressing information from multiple word tokens. Simply prompting to reason faster enables CoLaR to predict more informative latents.

## Abstract

Large Language Models (LLMs) achieve superior performance through Chain-of-Thought (CoT) reasoning, but these token-level reasoning chains are computationally expensive and inefficient. In this paper, we introduce Compressed Latent Reasoning (CoLaR), a novel framework that dynamically compresses reasoning processes in latent space through a two-stage training approach. First, during supervised fine-tuning, CoLaR extends beyond next-token prediction by incorporating an auxiliary next compressed embedding prediction objective. This process merges embeddings of consecutive tokens using a compression factor $c$ randomly sampled from a predefined range, and trains a specialized latent head to predict distributions of subsequent compressed embeddings. Second, we enhance CoLaR through reinforcement learning (RL) that leverages the latent head's non-deterministic nature to explore diverse reasoning paths and exploit more compact ones. This approach enables CoLaR to: i) **perform reasoning at a dense latent level** (i.e., silently), substantially reducing reasoning chain length, and ii) **dynamically adjust reasoning speed** at inference time by simply prompting the desired compression factor. Extensive experiments across four mathematical reasoning datasets demonstrate that CoLaR achieves $14.1\%$ higher accuracy than latent-based baseline methods at comparable compression ratios, and reduces reasoning chain length by $53.3\%$ with only $4.8\%$ performance degradation compared to explicit CoT method. Moreover, when applied to more challenging mathematical reasoning tasks, our RL-enhanced CoLaR demonstrates performance gains of up to $5.4\%$ while dramatically reducing latent reasoning chain length by $82.8\%$.

✉ Corresponding authors: Ruihua Song (rsong@ruc.edu.cn) and Jian Luan (luanjian@xiaomi.com).
Project page: `https://github.com/xiaomi-research/colar`.
*This Work was performed when Wenhui Tan was visiting Xiaomi as a research intern.

39th Conference on Neural Information Processing Systems (NeurIPS 2025).

# 1 Introduction

Large Language Models (LLMs) have demonstrated remarkable capabilities in mathematical reasoning, particularly when employing Chain-of-Thought (CoT) prompting techniques [29, 31]. Recent advances have further highlighted the potential of this approach when combined with reinforcement learning on extended reasoning sequences [16, 12, 28], revealing significant "aha-moments" in model performance. Despite these advances, a critical limitation persists: generating lengthy reasoning chains are computational costly, impeding efficiency and scalability. This inefficiency becomes particularly evident in real-world LLM applications, where extended reasoning chains create substantial server load, especially under high-concurrency conditions, underscoring the urgent need for more efficient reasoning methods.

Several approaches have emerged to address this computational challenge [26, 7]. One line of research focuses on enhancing efficiency at the token level, primarily by identifying and skipping less informative tokens [30], prompting models to generate more concise reasoning steps [32, 1], or dynamically terminating reasoning when the model exhibits high confidence in a trial answer [35]. While valuable, these methods continue to operate on sparse token-based representations. A more promising direction explores reasoning within the dense latent space. Initial efforts attempt to "internalize" reasoning knowledge by curriculum learning [5] or knowledge distillation [6]. Some works focus on the potential inside LLMs by looping or skipping some intermediate LLM layers [3, 2, 22] to realize efficient reasoning. Recent innovations have introduced auto-regressive prediction of latent representations for efficient reasoning. Coconut [13] proposes to gradually replace token-level reasoning with latent representations, while CODI [24] employs self-distillation to transfer CoT knowledge into latent reasoning processes. However, these methods primarily utilize **fixed-length reasoning chains**, resulting in suboptimal efficiency and limited adaptability. Furthermore, to the best of our knowledge, all these latent-based methods employ **deterministic latent reasoning processes**, overlooking the potential benefits of exploration-exploitation capability may bring about.

To overcome these limitations, we introduce Compressed Latent Reasoning (CoLaR), a novel framework that dynamically compresses LLM reasoning chains into latent space while preserving exploration-exploitation capabilities. Our approach utilizes an auxiliary *next compressed embedding prediction* task in supervised fine-tuning (SFT) stage. Specifically, at each **training** step, CoLaR first samples a random compression factor $c \in [1, c_{max}]$ and **merges the embeddings** of $c$ consecutive reasoning tokens using our Embedding Compress module. A Latent Head is then trained to **predict the next compressed embeddings** from the LLM's output hidden states, which is a fully parallelized process. **During inference**, CoLaR is capable to auto-regressively predict dense and informative latents with the Latent Head, and automatically determine when to terminate the reasoning process with LLM's Language Head. Rather than predicting deterministic values, the Latent Head outputs a probability distribution that produces diverse reasoning pathways for a same question input. Based on this, we further enhance CoLaR through post-training with Group Relative Policy Optimization (GRPO) reinforcement learning algorithm **(author?)** [23, 37], which enables CoLaR to **explore** correct latent reasoning paths with diverse outputs and **exploit** those shorter ones.

Our extensive evaluations on four grade-school level mathematical reasoning datasets (GSM8k [4], GSM8k-hard [9], SVAMP [20], and MultiArith [21]) demonstrate that CoLaR achieves a 14.1% ↑ improvement in accuracy compared to state-of-the-art baseline methods at comparable compression ratios. Furthermore, CoLaR reduces reasoning chain length by 53.3% ↓ with only a 4.8% ↓ performance degradation relative to explicit CoT. Finally, experiments on a more challenging dataset MATH [14] demonstrates the potential of CoLaR to reinforcement learning, gaining up to 5.36% ↑ accuracy while reducing the length of reasoning chain significantly by 82.8% ↓.

Our main contribution are three-fold:

- We introduce Compressed Latent Reasoning (CoLaR), a novel framework enabling dynamic-speed reasoning by auto-regressively predicting latent variables that encapsulate the compressed semantics of multiple word tokens. This allows for more efficient reasoning by operating in a compressed latent space.

- We design CoLaR with a probabilistic Latent Head and demonstrate the effectiveness of reinforcement learning on latent reasoning. This combination improves performance and reduces the length of reasoning chains by encouraging exploration of diverse reasoning paths and exploitation of the shorter ones.

- Extensive experiments show that CoLaR achieves a $14.1\%$ accuracy improvement over existing latent-based methods. Furthermore, reinforcement learning enhances performance by up to $5.36\%$ while simultaneously reducing reasoning chain dramatically length by $82.8\%$, demonstrating significant efficiency gains.

## 2 Related Work

### 2.1 Explicit LLM reasoning

Recent advances have demonstrated the strong reasoning capabilities of large language models (LLMs). The explicit reasoning approach, exemplified by Chain-of-Thought (CoT) reasoning [29], propose to prompt LLMs to generate intermediate reasoning steps through sequential token prediction before generating answers [36, 31, 27, 39, 17]. Subsequent work demonstrated that reinforcement learning techniques [23, 37, 38] can further improve performance on verifiable reasoning tasks like mathematical problem-solving, revealing an "aha-moment" that significantly boosts model's performances with longer thinking process [16, 12, 28].

However, the computational cost of processing these lengthy reasoning chains remains a significant bottleneck, motivating research into efficiency optimizations. Current solutions focus on identifying and skipping redundant tokens [30, 35] or encouraging more compact reasoning patterns using mathematical notations and programming language-like expressions [1, 32]. While these methods reduce reasoning chain length, they are fundamentally limited by the sequential token prediction.

### 2.2 Latent LLM reasoning

Latent reasoning approaches operate in a denser, continuous space, abstracting away from individual word tokens. These methods can be broadly categorized into three directions: knowledge internalization, architectural modifications, and auto-regressive latent reasoning.

The first direction, knowledge internalization, aims to embed reasoning capabilities directly into the model. iCoT-SI [5] attempts to internalize reasoning knowledge by progressively removing explicit reasoning steps during training, while Pause [10] proposes training models to reason within specialized token embeddings.

The second direction exploits the hierarchical structure of transformer layers, with proposals to dynamically skip or repeat layer computations [22, 2, 3, 25, 19]. These methods aim to reduce computational cost by selectively processing different layers.

The third direction, and most relevant to our work, explores auto-regressive latent reasoning [33]. Coconut [13] pioneered this approach by replacing token sampling with hidden state concatenation for breadth-first reasoning, while CODI [24] introduces an auto-regressive latent variable model through self-distillation. However, existing methods like Coconut and CODI are limited by their reliance on fixed-length reasoning chains due to the implicit nature of latent variables. Furthermore, they employ a deterministic approach to auto-regressive latent generation, neglecting the potential for exploration-exploitation strategies to further enhance model performance, particularly within a reinforcement learning framework.

In contrast, CoLaR advances **auto-regressive latent reasoning** by introducing a novel next compressed embedding objective. This allows the model to capture the semantics of multiple word tokens within a single latent variable and reason with dynamic chain lengths, leading to improved efficiency and performance. Moreover, CoLaR achieves significant performance gains and a dramatic reduction in latent reasoning length through reinforcement learning with a probabilistic latent prediction head.

## 3 Method

In this section, we introduce our task, notations, and our proposed method CoLaR. We focus on mathematical reasoning tasks using a dataset $D$, which consists of a question $\mathbf{t}_q = t_q^{1:L_q}$, a reasoning chain $\mathbf{t}_r = t_r^{1:L_r}$, and an answer $\mathbf{a}_r = t_a^{1:L_a}$, where $L_q$, $L_r$, and $L_a$ denote the respective token lengths. A representative example entry would be: *"Question: A set of 7 spoons costs \$21. If each*

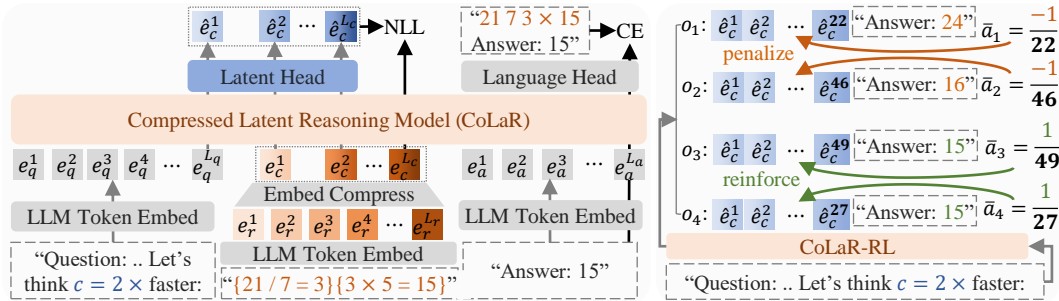

Figure 2: Our proposed method CoLaR consisting an LLM backbone and a Latent Head. During the **SFT stage (left)**, for each training step, CoLaR first compresses embeddings $\mathbf{e}_r$ of the original reasoning chain into compressed embeddings $\mathbf{e}_c$ with a compression factor $c$ randomly selected from the range $[1, c_{max}]$. Then, CoLaR is trained to predict: i) the compressed reasoning embeddings via the Latent Head, and ii) the compressed reasoning tokens and answer tokens through the Language Head. During the **RL stage (right)**, for every question input, CoLaR samples a group of $G$ outputs $o_{1:G}$ consisting of the latent reasoning chain and the predicted answer. We then calculate the relative rewards $a_{1:G}$ for each output, and the rewards are averaged on each token ($\bar{a}_i$), encouraging CoLaR to explore diverse latent reasoning pathways and exploit those more compact ones.

*spoon would be sold separately, how much would 5 spoons cost?", "Reasoning chain: « 21 / 7 = 3 » « 5 * 3 = 15 » <end>", and "Answer: 15".*

Given an LLM backbone $\mathcal{M}$, the input tokens are first mapped to embedding vectors $\mathbf{e}_q = e_q^{1:L_q}$, $\mathbf{e}_r = e_r^{1:L_r}$, and $\mathbf{e}_a = e_a^{1:L_a}$. These embeddings are processed by $\mathcal{M}$ to produce the hidden states of the final layer, denoted $\mathbf{h}_q = h_q^{1:L_q}$, $\mathbf{h}_r = h_r^{1:L_r}$, and $\mathbf{h}_a = h_a^{1:L_a}$. $\mathcal{M}$ then predicts distributions a.k.a. logits of next tokens using a Language Head.

To address this issue of lengthy reasoning chains, we propose compressing reasoning processes into a denser latent space, facilitating more efficient LLM reasoning. This requires our method to i) **compress** reasoning tokens into latent space and **understand** these dense representations, ii) **predict** subsequent dense latent representations and determine when to **terminate** reasoning, and iii) maintain the ability to **explore** diverse latent reasoning paths and **exploit** shorter latent solutions. CoLaR is designed with these three objectives in mind.

### 3.1 Reasoning token compression and understanding

As illustrated in Figure 2, the input to CoLaR can be represented as $\mathbf{e} = [\mathbf{e}_q, \mathbf{e}_c, \mathbf{e}_a]$, where $[\cdot, \cdot]$ denotes concatenation. Here, $\mathbf{e}_c$ represents the compressed embeddings derived from $\mathbf{e}_r$, the embeddings of the original reasoning steps, and the length of compressed embeddings $L_c = \lceil \frac{L_r}{c} \rceil$. To achieve a dynamic test-time compression factor $c$, we begin each training step by randomly sampling $c \in [1, r_{max}]$. For every $r$ consecutive reasoning token embeddings $e_r^{k:k+r}$, the Embedding Compress module generates a compressed embedding $e_c^k$.

A straightforward approach is to apply mean pooling directly to these embeddings. However, due to the high dimensionality of the embedding space (e.g., 2048 dimensions), embeddings from different tokens may be highly uncorrelated. Simply averaging these embeddings can distort the original distribution. For instance, consider two uncorrelated distributions $A \sim \mathcal{N}(\mu, \sigma^2)$ and $B \sim \mathcal{N}(\mu, \sigma^2)$; mean pooling would alter the original distribution to $\frac{A+B}{2} \sim \mathcal{N}(\mu, \frac{\sigma^2}{2})$, effectively scaling the variance by $\frac{1}{2}$. We found that, for most pre-trained LLMs, the distributions of embeddings are centered at $\mu \approx 0$. Thus, to prevent distortion of the original embedding distribution of LLMs, the Embedding Compress module only scales the sum of the $c$ embeddings by $\frac{1}{\sqrt{c}}$.

Intuitively, it could be difficult for LLMs to understand these compressed embeddings. A simple approach is to supervise $\mathcal{M}$ to predict answers with a language modeling loss, which enforces $\mathcal{M}$ to model answers with compressed embedding inputs. However, this objective provides supervision signals that are too sparse to converge to near-optimal performance. To address this issue, we train

CoLaR to predict the compressed reasoning tokens. Ideally, when using a compression factor $c$, CoLaR should be able to *read* and *predict* tokens in groups of $c$. This means that for each compressed embedding input, CoLaR should be trained to predict all $c$ corresponding tokens. To approximate this multi-label classification task using the single-label prediction capability of LLM's language model head, we randomly sample one token from each group of $c$ reasoning tokens $t_r^{k \times c:(k+1) \times c}$ as the ground-truth label. This approach trains the predicted logits to approximate a multimodal distribution that represents all potential tokens in each compressed group. This process could be formally represented as:

$$\mathcal{L}_{\text{comp}} = -\frac{1}{L_a + L_c} \sum_{i=1}^{L_a+L_c} \log p([\mathbf{t}_c, \mathbf{t}_a]^i | [\mathbf{e}_c, \mathbf{e}_a]^{1:i-1}, \mathbf{e}_q), \tag{1}$$

where $\mathbf{t}_c$ are sampled from $\mathbf{t}_r$.

## 3.2 Next compressed embedding prediction

To enable auto-regressive latent reasoning, we train a Latent Head $\mathcal{E}$ (analogous to the Language Head in LLMs) to predict the next compressed embedding, where $\mathcal{E}$ is a two-headed MLP. Given the current hidden states $h_c^i$ output by $\mathcal{M}$, the Latent Head $\mathcal{E}$ predicts both the mean $\mu_c^{i+1}$ and standard deviation $\sigma_c^{i+1}$ of the next embedding's distribution.

Unlike previous works that predict deterministic values—which limits exploration of alternative reasoning pathways—our approach generates a probabilistic distribution. During inference, we employ the re-parameterization trick to sample the next embedding: $\hat{e}_c^{i+1} = \hat{\mu}_c^{i+1} + \hat{\sigma}_c^{i+1} \epsilon$, where $\epsilon$ is random noise sampled from a standard Gaussian distribution $\mathcal{N}(0, 1)$.

The Latent Head $\mathcal{E}$ is primarily trained using the negative log-likelihood (NLL) loss. For a prediction at position $i$, this can be formulated as:

$$\mathcal{L}_{\text{latent}}(i) = -\log p(e_c^i \mid \hat{\mu}_c^i, \hat{\sigma}_c^i) = \frac{(e_c^i - \hat{\mu}_c^i)^2}{2\hat{\sigma}_c^i} + \log \hat{\sigma}_c^i \tag{2}$$

This probabilistic formulation enables the model to capture uncertainty in the latent reasoning process and allows for diverse reasoning pathways during generation. The total loss is computed by averaging over all positions in the compressed embedding sequence.

However, we empirically found that CoLaR with NLL loss tends to under-fit on *simpler* math reasoning datasets that require less exploration. To address this, we propose a soft-MSE loss that combines Mean Squared Error with an entropy regularization term:

$$\mathcal{L}_{\text{latent}}(i) = \underbrace{\mathbb{E}_\epsilon \left[ (\hat{\mu}_c^i + \hat{\sigma}_c^i \epsilon - e_c^i)^2 \right]}_{\text{MSE term}} - \alpha \underbrace{\left( \frac{1}{2} \log(2\pi e \left( \hat{\sigma}_c^i \right)^2) \right)}_{\text{entropy term}}, \tag{3}$$

where $\alpha$ is a positive hyperparameter that encourages the model to predict more diverse latents with larger $\hat{\sigma}$ values. This approach enables CoLaR to better fit simpler datasets while maintaining its exploration capability. We evaluate **both the two forms** of latent loss in our experiments. We sum up $\mathcal{L}_{comp}$ and $\mathcal{L}_{latent}$ as the final loss to optimize CoLaR in the SFT stage.

## 3.3 Exploration with reinforcement learning

The next-compressed embedding prediction training enables latent reasoning chains to *mimic* original chain of thoughts. However, this paradigm inevitably limits the performance of latent reasoning models to their CoT teachers. To further explore the potential of CoLaR, we apply a reinforcement learning stage to our proposed method.

With the Latent Head trained to predict distributions, we can sample diverse latent reasoning pathways and final answers for the same question $q$. We then apply Group Relative Policy Optimization (GRPO) algorithm [23] to reinforce correct reasoning chains and answers while penalizing incorrect ones.

Specifically, for each question $q$, GRPO first samples a group of outputs $\{o_1, o_2, \ldots, o_G\}$ from the old policy $\pi_{\theta_{\text{old}}}$, where $G$ is the group size. Each output $o_i$ consists of a latent reasoning chain and a final answer. Then, GRPO optimizes the policy $\pi_\theta$ by minimizing the following objective:

$$\mathcal{L}_{\text{GRPO}} = -\frac{1}{G} \sum_{i=1}^{G} \left( \min \left( \frac{\pi_\theta \left(o_i | q\right)}{\pi_{\theta_{\text{old}}} \left(o_i | q\right)} A_i, \text{clip} \left( \frac{\pi_\theta \left(o_i | q\right)}{\pi_{\theta_{\text{old}}} \left(o_i | q\right)}, 1 - \epsilon, 1 + \epsilon \right) A_i \right) \right), \qquad (4)$$

where $\epsilon$ is a hyperparameter, and $A_i$ is calculated as a group-normalized reward:

$$A_i = \frac{r_i - \text{mean} \left(r_1, r_2, \dots, r_G\right)}{\text{std} \left(r_1, r_2, \dots, r_G\right)}. \qquad (5)$$

We simply set $r_i$ to 1 when an answer is correctly predicted and to 0 otherwise. Following DAPO [37], we remove the KL-regularization term from original GRPO implementation for efficient training.

Notably, $\mathcal{L}_{GRPO}$ is calculated at the output level, i.e., across the entire latent reasoning chain and predicted answer, but is then **averaged** when applied to each latent/token. This design encourages CoLaR to balance exploration and exploitation. For instance, in Figure 2, although both $a_1 = a_2 = -1$, GRPO penalizes the latents/tokens in $o_1$ more, as there are fewer reasoning steps. This encourages CoLaR to think more deeply to **explore** correct reasoning paths. Likewise, the latents/tokens in $o_4$ are reinforced more as the reward is averaged less, which encourages CoLaR to **exploit** those more compact latent reasoning paths.

## 4 Experiments

In this section, we evaluate our proposed method CoLaR against strong baselines, analyze the contributions of different components, and explore the impact of key parameters.

### 4.1 Experimental setup

**Datasets and tasks.** Our method is mainly trained and evaluated on **GSM8k-Aug** [6], an augmented version of the Grade-School level Math reasoning dataset GSM8k [4]. GSM8k-Aug comprises approximately 385k training samples and 1k test samples. We also evaluate the trained methods on three out-of-domain math reasoning datasets: (1) **GSM-Hard** [9], a modified version of GSM8K with approximately 1k test samples featuring larger magnitude numbers, (2) **SVAMP** [20] and (3) **MultiArith** [21], two simpler math reasoning datasets with 1k and 600 test samples, respectively. Moreover, we train and evaluate our method on a more challenging dataset **MATH** [14], which consists of 7.5k training samples and 5k test samples, covering algebra, calculus, statistics, geometry, linear algebra and number theory. Following [13], we use two metrics: (1) Accuracy (Acc.), which measures the effectiveness of correctly predicting answers and (2) Reasoning chain length (# L), which measures efficiency by averaging the number of tokens/latents predicted in reasoning chains.

**Baseline methods.** We primarily compare against the following baselines: (1) **CoT** [29], which is fine-tuned on complete reasoning chains and answers, and performs token-level reasoning before predicting answers during inference; (2) **iCoT** [5], which internalizes reasoning knowledge by gradually removing reasoning steps, and directly predicts answers during inference; (3) **Coconut** [13], which is fine-tuned with a curriculum process to gradually replace token-level reasoning steps with latent reasoning steps, and performs six steps of latent reasoning before predicting answers; and (4) **Distill**, is our reproduced version of CODI [24] based on their implementation details as the code and model are not released. It self-distills token-level CoT into fixed-length latent reasoning steps, with an inference stage same to Coconut.

**Implementation details.** (1) Base model: unless otherwise specified, all experiments use a frozen Llama-3.2-1B-Instruct [11] backbone with a trainable LoRA module [15]. Following Coconut, all methods are initialized with weights from CoT-SFT to accelerate training. (2) Model checkpointing: for fair comparison, all models are trained for up to 50 epochs or 12 hours, whichever is reached first, and we choose the checkpoint that achieves the best accuracy on the validation set as the final model. (3) Hyper-parameters: we use the AdamW [18] optimizer with a fixed learning rate of 1e-4 and a weight decay of 1e-2 in SFT stage, and set the learning rate to 1e-6 in RL stage. We set $r_{max} = 5$ to train CoLaR. During inference, we configure the LLM generation with a temperature of 1 and top-p of 0.9. All SFT training processes are conducted a total batch size of 256. For more implementation details, please refer to Appendix Section A.

Table 1: Experiment results of baseline methods and CoLaR on four grade-school math reasoning datasets. We test the methods for five times with different random seeds to report the averaged number and 95% confidence interval ($\pm$) on accuracy (Acc. %) and reasoning chain length (# L). CoLaR-$c$ denotes a same CoLaR model tested with different compression factors $c$. For ablation methods (marked in gray), suffixes DL, OC, MP and NLL denote CoLaR with a Deterministic Latent head, training withOut Compressed reasoning chain in cross entropy labels, using Mean Pooling to compress embeddings, and training with NLL loss, respectively.

| | GSM8k-Aug | | GSM-Hard | | SVAMP | | MultiArith | | Average | |
| | Acc. | # L | Acc. | # L | Acc. | # L | Acc. | # L | Acc. | # L |
|---|---|---|---|---|---|---|---|---|---|---|
| CoT | $49.4_{\pm.72}$ | $25.6_{\pm.11}$ | $11.9_{\pm.16}$ | $34.2_{\pm.11}$ | $59.8_{\pm.29}$ | $12.1_{\pm.03}$ | $93.2_{\pm.49}$ | $13.7_{\pm.09}$ | 53.6 | 21.4 |
| iCoT | $19.8_{\pm.23}$ | $0.00_{\pm.00}$ | $3.87_{\pm.16}$ | $0.00_{\pm.00}$ | $36.4_{\pm.51}$ | $0.00_{\pm.00}$ | $38.2_{\pm.66}$ | $0.00_{\pm.00}$ | 24.6 | 0.00 |
| Coconut | $23.1_{\pm.28}$ | $6.00_{\pm.00}$ | $5.49_{\pm.33}$ | $6.00_{\pm.00}$ | $40.7_{\pm.65}$ | $6.00_{\pm.00}$ | $41.1_{\pm.24}$ | $6.00_{\pm.00}$ | 27.6 | 6.00 |
| Distill | $13.3_{\pm.62}$ | $6.00_{\pm.00}$ | $2.97_{\pm.24}$ | $6.00_{\pm.00}$ | $21.7_{\pm.73}$ | $6.00_{\pm.00}$ | $19.2_{\pm.83}$ | $6.00_{\pm.00}$ | 14.3 | 6.00 |
| CoLaR-5 | $26.8_{\pm.17}$ | $5.57_{\pm.02}$ | $5.87_{\pm.10}$ | $6.53_{\pm.01}$ | $48.4_{\pm.45}$ | $2.95_{\pm.02}$ | $86.4_{\pm.35}$ | $3.21_{\pm.01}$ | 41.7 | 4.57 |
| - DL | $26.7_{\pm.11}$ | $5.74_{\pm.01}$ | $5.53_{\pm.11}$ | $8.20_{\pm.04}$ | $48.3_{\pm.05}$ | $2.90_{\pm.01}$ | $84.5_{\pm.19}$ | $3.22_{\pm.01}$ | 41.3 | 5.02 |
| - OC | $24.8_{\pm.27}$ | $5.14_{\pm.12}$ | $6.46_{\pm.11}$ | $5.49_{\pm.06}$ | $46.5_{\pm.18}$ | $2.85_{\pm.01}$ | $85.9_{\pm.22}$ | $3.13_{\pm.01}$ | 40.1 | 4.15 |
| - MP | $20.6_{\pm.22}$ | $5.61_{\pm.02}$ | $4.20_{\pm.07}$ | $6.18_{\pm.02}$ | $47.7_{\pm.41}$ | $2.96_{\pm.01}$ | $80.7_{\pm.59}$ | $3.20_{\pm.01}$ | 38.3 | 4.49 |
| - NLL | $20.3_{\pm.64}$ | $5.99_{\pm.06}$ | $4.52_{\pm.39}$ | $16.6_{\pm.25}$ | $43.9_{\pm.43}$ | $3.06_{\pm.03}$ | $81.6_{\pm.23}$ | $3.20_{\pm.02}$ | 37.6 | 8.01 |
| CoLaR-2 | $40.1_{\pm.20}$ | $12.7_{\pm.02}$ | $9.08_{\pm.03}$ | $14.0_{\pm.07}$ | $54.9_{\pm.20}$ | $6.11_{\pm.01}$ | $91.3_{\pm.12}$ | $7.35_{\pm.01}$ | 48.8 | 10.0 |
| - DL | $39.7_{\pm.18}$ | $12.8_{\pm.01}$ | $8.84_{\pm.06}$ | $17.2_{\pm.09}$ | $54.3_{\pm.23}$ | $6.10_{\pm.01}$ | $90.1_{\pm.17}$ | $7.46_{\pm.01}$ | 48.2 | 10.9 |
| - OC | $39.1_{\pm.33}$ | $12.3_{\pm.04}$ | $8.96_{\pm.01}$ | $16.9_{\pm.13}$ | $54.7_{\pm.18}$ | $6.08_{\pm.02}$ | $90.1_{\pm.25}$ | $7.36_{\pm.01}$ | 48.2 | 10.6 |
| - MP | $36.9_{\pm.30}$ | $12.4_{\pm.02}$ | $8.46_{\pm.19}$ | $12.0_{\pm.05}$ | $54.1_{\pm.42}$ | $6.14_{\pm.01}$ | $86.8_{\pm.20}$ | $7.43_{\pm.01}$ | 46.6 | 9.49 |
| - NLL | $32.3_{\pm.51}$ | $12.2_{\pm.04}$ | $7.57_{\pm.16}$ | $16.6_{\pm.25}$ | $51.0_{\pm.24}$ | $5.50_{\pm.03}$ | $88.3_{\pm.41}$ | $7.09_{\pm.02}$ | 44.8 | 10.3 |

## 4.2 Comparison to baseline methods on GSM datasets

Table 1 presents a comparison of CoLaR against state-of-the-art baseline methods on four grade-school level math reasoning datasets. CoLaR demonstrates consistent performance gains over existing latent-based reasoning approaches. Notably, CoLaR with a test-time compression factor of 5 (CoLaR-5) achieves a 14.1% improvement in average accuracy compared to Coconut, and does so with fewer reasoning steps (4.57 vs. 6.00). This advantage stems from our effective next-compressed embedding prediction objective, which efficiently compresses the reasoning process into compact and informative latent variables. This allows for superior performance while maintaining a high compression ratio.

Leveraging the dynamic compression design, we evaluated the same trained model with a different test-time compression factor of 2 (CoLaR-2). The resulting accuracy of 48.8% represents only a 4.8% decrease compared to explicit CoT, but with a significant 53% reduction in reasoning chain length.

Furthermore, CoLaR exhibits robust out-of-domain generalization capabilities compared to other latent-based baselines, particularly on the MultiArith dataset, where CoLaR shows minimal performance degradation compared to CoT, while other latent-based methods suffer significant drops.

## 4.3 Ablation studies of on GSM datasets

We conduct ablation studies with four experimental settings on the GSM datasets, with results illustrated in the gray areas of Table 1. Three key findings emerge:

(1) **Simple math questions require balanced exploration capability.** Comparing CoLaR with CoLaR-DL (trained with a deterministic latent head) and CoLaR-NLL (trained with NLL loss), we find that a deterministic latent head fits well on simple math datasets but lacks test-time exploration potential, leading to suboptimal performance. Conversely, training CoLaR with NLL loss introduces excessive randomness, resulting in poor fit on the training data and worse overall performance.

(2) **Dense reasoning supervision signals are crucial.** When comparing CoLaR with CoLaR-OC, where we remove the tokens of compressed reasoning chain $\mathbf{t_c}$ and use only the final answer tokens $\mathbf{t_a}$ as the language modeling supervision signal, performance degrades by 1.6% and 0.6% at compression factors $c = 5$ and $c = 2$, respectively. This confirms the importance of dense supervision signals when training latent-based reasoning methods.

(3) **Latents under different compression factors should share the same space.** CoLaR-MP, which applies Mean Pooling on the compression embeddings, shows 3.4% and 2.2% performance degradation compared to our method. This decline is primarily attributed to distribution shifts caused by the compression process, which introduces confusion during model training.

Table 2: Experimental results on the challenging MATH dataset. We evaluate our proposed method CoLaR on two base models and three settings: -DL denotes using a Deterministic Latent head, -NLL denotes CoLaR trained with NLL Loss as $\mathcal{L}_{\text{latent}}$, which is our main method, and - /w GRPO denotes the post-trained CoLaR-NLL with GRPO reinforcement learning process. We calculate the performance gain between CoLaR-NLL and CoLaR-NLL-RL to highlight the effectiveness of reinforcement learning. Compression factor $c$ and # $\text{L}_{max}$ are set to 2 and 128, respectively.

|  | DeepSeek-R1-Distill-Qwen-1.5B | | Llama-3.2-1B-Instruct | |
|  | Acc. | #L | Acc. | #L |
|---|---|---|---|---|
| CoT | $23.5_{\pm.29}$ | $209_{\pm1.6}$ | $9.71_{\pm.33}$ | $210_{\pm1.4}$ |
| CoLaR-DL | $9.04_{\pm.12}$ | $99.4_{\pm.25}$ | $3.07_{\pm.28}$ | $134_{\pm.46}$ |
| CoLaR-NLL | $8.94_{\pm.21}$ | $56.8_{\pm.14}$ | $5.28_{\pm.16}$ | $83.1_{\pm.52}$ |
| CoLaR-NLL-RL | $14.3_{\pm.25}$ (5.36% ↑) | $9.79_{\pm.40}$ (82.8% ↓) | $7.08_{\pm.07}$ (1.80% ↑) | $16.1_{\pm.14}$ (80.6% ↓) |
| - w/o average | $13.8_{\pm.14}$ | $128_{\pm.00}$ | $0.00_{\pm.00}$ | $128.0_{\pm.00}$ |

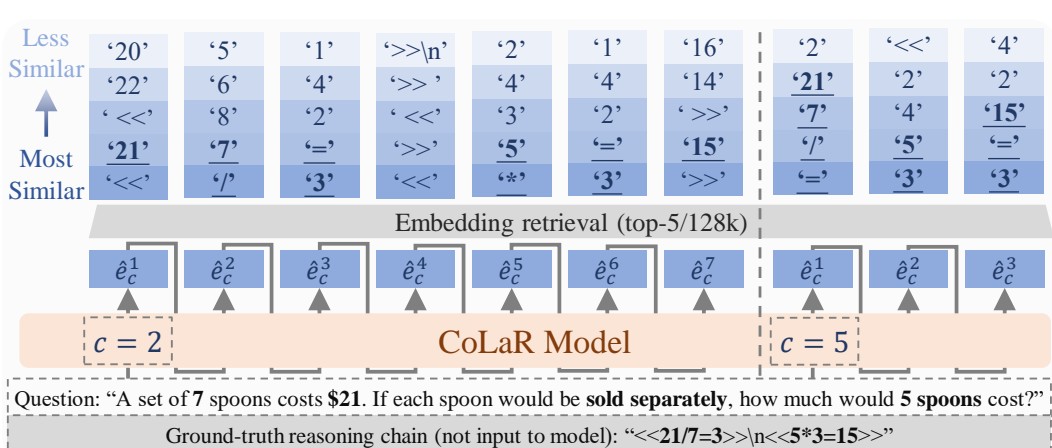

Figure 3: A case study on the GSM-8k validation set. We set the compression factor $c$ to 2 and 5, which produce two latent reasoning chains in length 7 and 3, respectively. We then retrieve tokens with the predicted latents by embedding cosine similarity, and underscore those informative tokens.

## 4.4 Reinforcement learning results on the MATH dataset

We train and evaluate CoLaR with RL on the challenging MATH dataset, using two base models [12, 34, 11]. The results are presented in Table 2. Our analysis of the results yields three key conclusions:

(1) **Exploration is crucial for difficult problems.** The deterministic latent reasoning process of CoLaR-DL exhibits accuracy comparable to or worse than CoLaR-NLL despite longer reasoning chains. This suggests that challenging math problems necessitate exploration of multiple potential solutions, rather than deterministic, step-by-step reasoning. Furthermore, post-training CoLaR-NLL with RL yields significant gains, achieving up to 5.36% higher accuracy and an 82.8% reduction in reasoning length. This highlights the potential of RL and the importance of balancing exploration and exploitation for latent reasoning models.

(2) **Averaged rewards promote exploitation.** When training without this averaging (i.e., simply dividing the loss by a constant to normalize the loss scale), we observed that while Qwen-1.5B exhibited a performance increase (from 8.94% to 13.8%) similar to averaging the loss, the reasoning length rapidly converged to the pre-defined upper limit. Moreover, Llama-1B's performance tend towards collapse. This suggests the averaged design encourages CoLaR to exploit more efficient reasoning pathways.

(3) **Base model quality impacts RL effectiveness.** Supervised fine-tuning on CoT resulted in varying performance across the two base LLMs. Meanwhile, CoLaR also demonstrates a significantly larger performance gain on RL when using the higher-quality Qwen-1.5B compared to Llama-1B. This

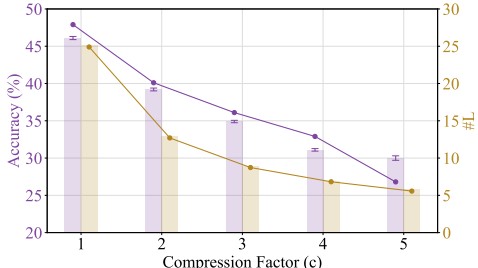 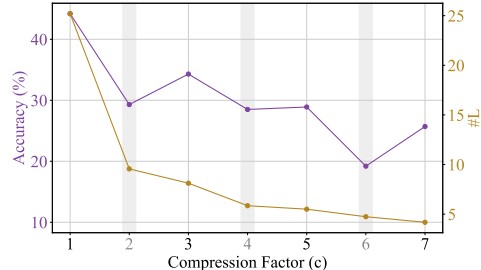

Figure 4: Accuracy and reasoning chain length (# L) of CoLaR on GSM8k dataset when trained with random $c \in [1, 5]$ (the **lines**) or trained solely on specific $c$ (the **bars**).

Figure 5: Accuracy and reasoning chain length (# L) of CoLaR on GSM8k dataset when trained with $c \in \{1, 3, 5, 7\}$ and tested with extra $c \in \{2, 4, 6\}$ (under gray bars).

Table 3: Experimental results of CoLaR-2 and TokenSkip on GSM-8k, MATH-500, and GPQA.

| Acc. (%)/#L | CoT-1B | CoLaR-1B | CoT-8B | CoLaR-8B | CoLaR-8B-RL | TokenSkip-8B |
|---|---|---|---|---|---|---|
| GSM-8k | 47.5/92.6 | 40.4/49.2 | 76.5/93.6 | 70.8/47.3 | 71.9/13.3 | 78.2/113 |
| MATH-500 | 28.6/176 | 19.8/84.6 | 54.6/168 | 45.8/67.7 | 52.4/17.6 | 40.2/292 |
| GPQA | 26.4/232 | 26.4/84.1 | 35.7/216 | 32.4/101 | 37.5/66.7 | - |

observation aligns with the findings of [8], which suggests that RL substantially **activates inherent reasoning capabilities**, indicating the importance of base model quality.

We also observed that during the RL training process, CoLaR tends to *think longer* initially with a rapid rise in accuracy, followed by a phase of *thinking shorter* accompanied by a more stable increase in accuracy, aligning with the discussion in Section 3.3. Due to space constraints, the detailed training curves are provided in Appendix C.

## 4.5 Scaling CoLaR on larger base model

We scale CoLaR to the 1-Billion and 8-Billion parameter Llama checkpoints to verify that its compression and accuracy gains grow with model capacity. The 8B model also matches the backbone used by TokenSkip, allowing a direct comparison under its published protocol. Experiments are run on GSM-8k, MATH-500 which are slightly different to Section 4.1 to align with TokenSkip's settings) and the out-of-domain GPQA benchmark (chemistry, biology and physics). The results are shown in Table 3. Moving from 1 B to 8 B parameters raises CoLaR's accuracy on every dataset while keeping the reasoning chain roughly half the length of the CoT teacher. Reinforcement learning delivers further improvements: +1.1% on GSM-8k, +6.6% on MATH-500 and +5.1% on GPQA, again with large additional compression. Most notably, CoLaR-8B-RL reaches 37.5% on GPQA, surpassing its 35.7% CoT teacher and shortening the reasoning chain by 69%, confirming that latent RL remains effective as models grow.

## 4.6 Interpreting chains of latent thoughts

To quantatively illustate the effectiveness of our proposed CoLaR, and make the latent reasoning process more transparent, we conduct a case study of CoLaR on the GSM8k validation set. The results are illlustrated in Figure 3.

As latent variables are fundamentally scaled sum of word token embeddings, we could directly calculate the cosine simility between latent variables and the enitre LLM embedding matrix:

- When prompted with compression factor $c = 2$, CoLaR auto-regressively produces seven latent variables, and then automatically terminates reasoning process. By calculating similarity scores, each latent variable is capable of retrieving meaningful words such as "«",

"*21*", and the entire chain of latent thoughts could be interpretered as "*«21/7=3» «5*3=15»*", which exactly matches the correct calculation process.

- When prompted with higher compression factor $c = 5$, less informative tokens such as "«" are ignored, demonstrating both the effectiveness and efficiency of our proposed dynamic compression mechanism.

### 4.7  Analyses on dynamic compression factors

We investigate the generalization capability of CoLaR across different compression factors $c$. Two key findings emerge from our analyses:

First, as illustrated in Figure 4, for each test-time compression factor (except from $c = 5$), Co-LaR trained with random $c \in [1, 5]$ consistently outperforms models trained on a single compression factor. These results demonstrate that exposure to diverse training-time compression factors produces complementary benefits for generalization. For example, training with $c = 2$ also improves the performance of testing with $c = 4$, highlighting the effectiveness of our dynamic training process.

Second, as shown in Figure 5, we train CoLaR with $c \in \{1, 3, 5, 7\}$ and evaluate it with previously unseen compression factors $c \in \{2, 4, 6\}$. We find that CoLaR successfully generalizes to these unseen compression factors, maintaining expected actual compression rates. Moreover, though worse in absolute values, the slope of the performance curve on out-of-domain compression factors closely resembles that of in-domain factors, suggesting robust interpolation capabilities.

## 5  Limitations

While CoLaR demonstrates superior effectiveness and efficiency in latent reasoning, we acknowledge two important limitations: (1) Though CoLaR surpassing its CoT teacher on GPQA, on the remaining benchmarks, the overall performance of CoLaR currently *approximates* explicit CoT reasoning without surpassing it. (2) We observe that CoLaR struggles to generalize to non-integer compression factors (e.g., $c = 1.5$) or to values greater than the maximum training compression factor $r_{max}$. This limitation stems primarily from the discrete tokenization constraints inherent to large language models, which restrict the continuous representation of compression factors. (3) Beyond technical limitations, our work on enhancing reasoning capabilities in LLMs has significant societal implications. On the positive side, CoLaR could significantly boost the efficiency of existing LLM services. However, potential negative impacts include the risk of amplifying existing biases in reasoning processes and possible misuse for generating more convincing misinformation. To mitigate these risks, we recommend careful monitoring of downstream applications.

## 6  Conclusion

In this paper, we introduce Compressed Latent Reasoning (CoLaR), a framework that dynamically compresses LLM reasoning chains into latent space while maintaining exploration-exploitation capabilities. Our method centers on three key innovations: (1) compressed latent reasoning through an auxiliary next compressed embedding prediction task that encapsulates the semantics of multiple tokens, (2) dynamic training and inference with variable compression factors that allows for flexible reasoning chain lengths and fully parallelized processing, and (3) a probabilistic latent head for reinforcement learning that enables exploration of diverse reasoning pathways for higher accuracy while exploiting shorter reasoning chains for efficiency. Our experimental results demonstrate that CoLaR achieves a 14.1% improvement in accuracy compared to state-of-the-art latent-based reasoning methods, while reducing reasoning chain length by 53.3% with only a 4.8% performance degradation relative to explicit CoT. On the challenging MATH dataset, reinforcement learning techniques further boost CoLaR's performance by 5.36% while dramatically reducing reasoning chain length by 82.8%. Future work will focus on addressing non-integer compression factors, exploring more sophisticated reinforcement learning approaches, and extending our dynamic compression mechanism to more diverse reasoning tasks beyond mathematics.

**Acknowledgements.** This work is supported by the Beijing Natural Science Foundation (L233008) and Xiaomi Inc. We acknowledge the anonymous reviewers for their helpful comments.

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

# A   More implementation details

In this section, we provide comprehensive details regarding our model architecture, training hyperparameters, and dataset specifications.

**Model hyperparameters.** For our experiments, we employ either a frozen LLama-3.2-1B-Instruct or DeepSeek-distill-Qwen-1.5B as our LLM backbone, augmented with a tunable LoRA module. All LoRA modules are configured with $\alpha = 32$ and $r = 128$ consistently across all experiments. Our method incorporates a Latent Head, implemented as a three-layered MLP with hidden dimensions corresponding to the LLM backbone's dimension ($d = 2048$).

**Training hyperparameters.** We utilize the AdamW optimizer with a weight decay of 1e-2 throughout our experiments. The learning rate is set at 1e-4 for supervised fine-tuning (SFT) and 1e-6 for reinforcement learning (RL). For SFT experiments, we leverage Distributed Data Parallel across eight A100 GPUs with a total batch size of 256. The RL experiments are conducted on a single A100 GPU with a rollout batch size of 8, optimizer step batch size of 4, group size $G$ of 8, and clip $\epsilon$ of 0.2. To ensure reproducibility, we fix the random seed of all libraries (Python, CUDA, PyTorch, and NumPy) to 0 for training processes. For evaluation, we use five distinct runs with random seeds sequentially set from 0 to 4.

Notably, when training the Latent Head, we normalize the target (i.e., the ground-truth compressed embeddings) to ensure training stability. This normalization is implemented by dividing the target by the standard deviation $\sigma_e$ of the embeddings. Since the embedding distributions are already centered at approximately zero ($\mu \approx 0$), we do not apply any shift during normalization. During inference, we multiply the predicted embeddings by the standard deviation to rescale them to match the LLM's original embedding distribution. These statistics can be either learned during training or calculated in advance; we opt for the latter approach for simplicity. We observe model-specific values, with $\sigma_e \approx 0.02$ for Llama-3.2-1B-Instruct and $\sigma_e \approx 0.03$ for Qwen-1.5B. This normalization process is critical for maintaining numerical stability while preserving the relative relationships between embedding dimensions.

**Dataset information.** We evaluate our method on four grade-school mathematics datasets—GSM8K-Aug, GSM8k-Hard, SVAMP, and MultiArith—as well as the more challenging MATH dataset for advanced mathematical reasoning. Since the original MATH dataset does not provide an official validation set, we randomly shuffle the training set and allocate 10% of the samples for validation purposes.

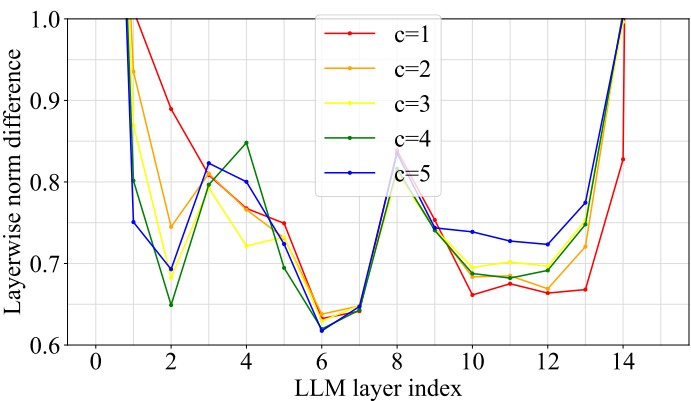

Figure 6: Layer-wise norm differences from CoLaR-1 to CoLaR-5.

## B    Layer-wise analyses on compression factors

We further investigate how the compression factor $c$ influences activation patterns across LLM layers, with results shown in Figure 6. Specifically, we tested CoLaR on the same sample as in Section 4.6 with compression factors ranging from 1 to 5, calculating the relative activation norm differences between consecutive LLM layers.

Our analysis reveals distinct patterns across different network depths:

- **Shallow layers** (0-3, near input): CoLaR shows higher **activation** on smaller compression factors with more pronounced layer-wise changes in magnitude.
- **Intermediate layers** (3-9): Models with different compression factors exhibit similar behavior.
- **Deeper layers** (9-15, near output): Higher compression factors maintain stronger activation patterns.

This phenomenon can be explained as follows: when predicting less informative tokens (e.g., "«") with lower compression factors (especially with $c = 1$, which uses no compression), the model requires minimal "thinking" and can determine the next token using primarily shallow layers. Consequently, computation in deeper layers is largely *underutilized*.

In contrast, higher compression factors enable CoLaR to process information more **densely**, with each latent representation carrying richer semantic content. This requires deeper layers to remain actively engaged in analyzing the condensed information and predicting subsequent compressed latents, thereby making more efficient use of the model's computational capacity. These findings align with observations from previous work on internal thinking processes in transformer models [2, 22].

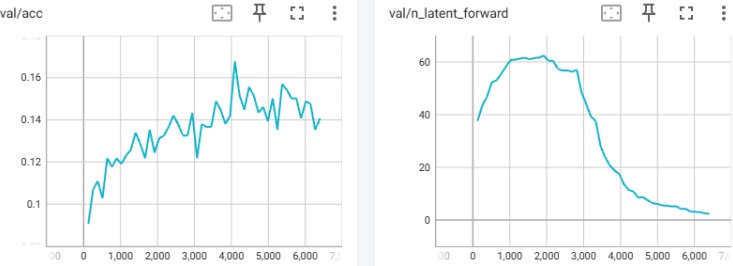

Figure 7: The validation accuracy and latent reasoning chain length curve on MATH dataset.

## C    RL training curves

Figure 7 presents the training curves from our reinforcement learning phase. The accuracy on the validation set exhibits a distinct three-phase pattern. In the initial exploration phase, accuracy increases rapidly from 9% to 14%, accompanied by an expansion of latent reasoning steps from 40 to 60. During this phase, the GRPO algorithm primarily encourages CoLaR to explore more extensively to discover correct reasoning pathways.

In the subsequent exploitation phase, validation accuracy fluctuates between 14% and 16%, while the latent reasoning length decreases from 60 to 20. With the per-token averaged reward/loss, the GRPO algorithm reinforces CoLaR to exploit shorter yet effective reasoning pathways.

Finally, as CoLaR begins to overfit, our early-stopping strategy is triggered to preserve the best-performing checkpoint at approximately 4k steps.

## D    Scaling properties of CoLaR

Figure 8 illustrates the performance characteristics of CoLaR when implemented with foundation models of varying parameter counts, ranging from 1 billion to 8 billion parameters. Our results

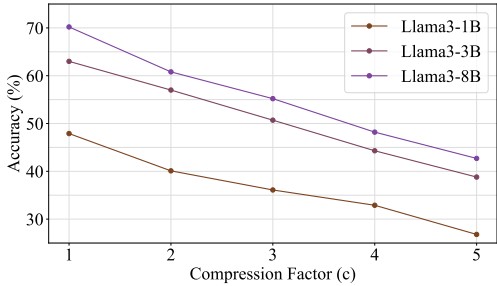

Figure 8: Performance of CoLaR when implemented with base LLMs ranging from 1B to 8B parameters.

demonstrate that CoLaR follows established neural scaling laws, with performance improvements correlating predictably with increases in the underlying model size. This consistent scaling behavior suggests that the benefits of our approach extend proportionally across different model scales, indicating CoLaR's architectural effectiveness is not limited to specific parameter regimes.

