# OpenReview forum: "Think Silently, Think Fast: Dynamic Latent Compression of LLM Reasoning Chains"
_NeurIPS.cc/2025/Conference — NeurIPS 2025 poster_

### Official Review · Reviewer_MsBS · 2025-06-23

**Clarity:** 3
**Significance:** 2
**Originality:** 2
**Rating:** 4
**Confidence:** 3

**Summary:**

The paper introduces Compressed Latent Reasoning (CoLaR), a novel framework for enhancing the efficiency of reasoning in LLMs by dynamically compressing reasoning chains into latent space. The approach leverages a two-stage training process: SFT with an auxiliary objective for predicting compressed embeddings, followed by RL to explore diverse reasoning paths. The results demonstrate significant improvements in accuracy and substantial reductions in reasoning chain length across various mathematical reasoning datasets.

**Questions:**

Please refer to "Weaknesses"

**Ethical Concerns:**

["NO or VERY MINOR ethics concerns only"]

**Final Justification:**

I appreciate the authors' reply. As most of my concerns have been addressed, I have adjusted my rating accordingly.

**Limitations:**

yes

**Quality:**

2

**Strengths And Weaknesses:**

Strengths
1. CoLaR introduces a novel method for compressing reasoning chains, allowing for faster and more efficient inference.
2. The ability to adjust compression factors at inference time demonstrates flexibility and adaptability
3. Extensive evaluations across multiple datasets demonstrate consistent performance gains.

Weaknesses
1. The evaluation primarily focuses on math problem solving  tasks, lacking assessments on more complex challenges.  Table 1 includes two version of GSM8k, which may confuse readers as to why two versions of the same dataset were chosen over other diverse datasets.
2. Testing is only conducted on small models, including 1B and 1.5B model. CoT reasoning typically benefits more from larger models, which may limit the applicability of findings.
3. The paper does not compare CoLaR with other concurrent reasoning compression methods, such as Tokenskip [1], LightThinker [2] or similar approaches that also optimize reasoning efficiency.
4. The selection of GRPO over other reinforcement learning algorithms is not justified, leaving questions about this choice.
5. CoLaR struggles to generalize to non-integer compression factors (e.g.,c=1.5) due to discrete tokenization constraints, which restricts its flexibility. Besides, CoLaR’s intermediate reasoning steps, compressed into latent space, are difficult for humans to interpret, as they lack the explicit, token-level reasoning provided by traditional CoT methods. This may opacity reduces the explainability of the model’s reasoning process.




Reference

[1] Xia, Heming, et al. "Tokenskip: Controllable chain-of-thought compression in llms." arXiv preprint arXiv:2502.12067 (2025).
[2] Zhang, Jintian, et al. "Lightthinker: Thinking step-by-step compression." arXiv preprint arXiv:2502.15589 (2025).

---

> ### Author Rebuttal · Authors · 2025-07-31
>
> Dear Reviewer #MsBS,
>
> Thank you for acknowledging CoLaR’s novelty and empirical breadth, and for enumerating the remaining gaps in domain coverage, scale, dataset, comparative baselines, algorithmic justification, and explainability. We address each concern below.
>
> **W1.1: Clarification of GSM8k.** Thank you for pointing this out. There are three versions of GSM8k: 1) GSM8k, the original one, where there are 8k samples with natural language reasoning chains, and 2) GSM8k-Aug, the one we used for training, which has extended training samples (\~385k) and condensed reasoning chains, and 3) GSM8k-Hard, the one we used for OOD testing, where the numbers are in larger range and harder to compute. The test set of GSM8k and GSM8k-Aug are the same. We will clarify this in our final revision.
>
> **W1.2 & W2 & W3: Scaling-up, new baseline, and non-math reasoning tasks.** Yes, CoLaR is capable to scale up to large foundation models, generalize to non-mathematical reasoning domains, and CoLaR surpasses TokenSkip on MATH dataset with fewer reasoning steps.
>
> Appendix E (Figure 8) already shows that CoLaR on GSM8k climbs from 26\~40% (1B) to 43\~61% (8B) while retaining comparable compression ratios.
>
> To extend this evidence, we ran CoLaR-2 on Llama-3.2-1B and Llama-3.1-8B across math and GPQA dataset, which consists high-quality difficulty question on physics, chemistry, and biology:
>
> (The reported results of GSM and MATH are slightly different from Table 1 of our manuscript. This is because we are using a different version datasets, GSM8k instead of GSM8k-Aug, to align with TokenSkip. We are unable to directly compare our CoLaR with LightThinker, which mainly focuses on replacing already-generated cot tokens with dense representations to reduce memory peak.)
>
> |       |   CoT-1B  |  CoLaR-1B |   |   CoT-8B  |  CoLaR-8B | CoLaR-8B-RL | TokenSkip-8B |
> |-------|:---------:|:---------:|---|:---------:|:---------:|:-----------:|:------------:|
> | GSM8k | 47.5/92.6 | 40.4/49.2 |   | 76.5/93.6 | 70.8/47.3 |  71.9/13.3  |   78.2/113   |
> | MATH  |  28.6/176 | 19.8/84.6 |   |  54.6/168 | 45.8/67.7 |  52.4/17.6  |   40.2/292   |
> | GPQA  |  26.4/232 | 26.4/84.1 |   |  35.7/216 |  32.4/101 |  37.5/66.7  |       -      |
>
> Four patterns emerge:
> - First, bigger backbones improve absolute accuracy for both CoT and CoLaR, yet CoLaR’s compression ratio remains stable.
> - Second, RL post-training raises CoLaR-8B above CoT on GPQA (37.5 vs 35.7) while shrinking reasoning chains by 70%.
> - Third, the gains generalize beyond mathematics, confirming that our paradigm scales and broadens with model size.
> - Finally, CoLaR-8B surpasses TokenSkip by a large margin on SFT and RL version with fewer reasoning steps on MATH dataset.
>
> **W4: Reasons to use GRPO.** Thank you for this question. GRPO is selected because the task is a canonical exploration–exploitation dilemma: for each question a combinatoric space of latent reasoning trajectories exists, yet only the subset that yields the correct answer is admissible.
>
> GRPO instantiates this balance explicitly: it first explores by sampling a group of complete latent chains, then exploits by reinforcing those that are both correct and minimal in length.
>
> Moreover, in contrast to PPO, GRPO dispenses with a learned value function and instead relies on the verifiable reward alone, reducing variance and computational overhead while preserving principled policy optimization.
>
> We will add relevant discussion to Section 3.3 in our final revision.
>
> **W5.1: Generalization to non-integer compression factors.** As acknowledged in the Limitations Section, CoLaR’s reliance on discrete prompting precludes fine-grained, non-integer compression ratios.
>
> We propose to address this limitation via a stochastic training strategy that blurs compression factor boundaries: at each forward pass, we randomly merge either one or two consecutive tokens, thereby inducing an effective compression factor that fluctuates around 1.5.
>
> This strategy approximates continuous control while remaining fully compatible with the model’s discrete tokenization, and we will empirically validate its efficacy in our future work.
>
> **W5.2: Explainability.** We thank the reviewer for this concern. Appendix B, Figure 5 demonstrates that CoLaR’s latent variables—compressed token embeddings—can be faithfully decoded: a simple cosine-similarity search against the token embedding matrix reliably recovers human-readable fragments such as “21 / 7 = 3 × 5 = 15.”
>
> More broadly, recent studies [1] reveal that explicit CoT steps can diverge from the model’s actual decision process or harbor silent biases; verbalization is therefore not a guarantee of transparency. Retrieving embeddings from CoLaR offers the same level of interpretability as CoT, while being more concise.
>
> Finally, we argue that human thought itself resides in a latent space. Russian distinguishes голубой (“sky-blue”) from синий (“ocean-blue”)—concepts with no exact English counterpart—yet speakers share the same perceptual embedding in their minds. An old adage captures this: “It can be grasped, yet cannot be spoken.” Echoing Wittgenstein, we believe much of human reasoning unfolds in a non-linguistic, latent form; CoLaR merely gives us direct access to that space and lets us translate it back into words when necessary.
>
> [1] Barez, Fazl, et al. "Chain-of-thought is not explainability." Preprint, alphaXiv (2025): v2.
>
> Your insightful remarks have been invaluable to the refinement of our work; we stand ready to address any additional concerns you may have.
>
> Authors of CoLaR #12942

---

> > ### Author Response · Authors · 2025-08-04
> > **Additional Clarification on Scaling Experiment Format**
> >
> > Dear Reviewer #MsBS,
> >
> > We'd like to briefly clarify the scaling experiment results: the values like "47.5/92.6" are reported in **"accuracy/reasoning chain length"** format. We hope this additional clarification helps address any potential confusion.
> >
> > We truly value your expertise and appreciate your time in reviewing our work, and we'd be grateful if you could take our updated response into consideration when finalizing your evaluation. If there are any remaining points you'd like us to elaborate on, please don't hesitate to let us know.
> >
> > Thanks again for your consideration!
> >
> > Authors of CoLaR #12942

---

> > > ### Comment · Reviewer_MsBS · 2025-08-08
> > >
> > > I appreciate the authors' reply. As most of my concerns have been addressed, I have adjusted my rating accordingly.

---

> > > > ### Author Response · Authors · 2025-08-08
> > > > **Response to Reviewer #MsBS**
> > > >
> > > > Dear Reviewer #MsBS,
> > > >
> > > > Thank you for your thoughtful comments and for revising your score. In the final revision, we will:
> > > > - expand the motivation for adopting GRPO in Section 3.3,
> > > > - clarify the GSM dataset series in Section 4.1,
> > > > - add scaling experiments and additional evaluation datasets in Section 4.2, and
> > > > - strengthen the discussion on explainability in the main text.
> > > >
> > > > Best regards,
> > > >
> > > > Authors of CoLaR #12942

---

### Official Review · Reviewer_nEhT · 2025-07-02

**Clarity:** 2
**Significance:** 3
**Originality:** 3
**Rating:** 4
**Confidence:** 3

**Summary:**

This paper proposes a learnable framework for semantic compression in latent space, aiming to reduce the length of reasoning chains. Although the performance slightly drops in certain cases, the method significantly shortens the Chain-of-Thought (CoT) reasoning steps.

**Questions:**

The paper emphasizes that the method reduces reasoning length. However, during inference, how does the model determine when to stop generating latent steps? Does it rely on checking, at every step, whether the Language Head produces a specific termination token? If so, wouldn’t this introduce additional computational overhead?

**Ethical Concerns:**

["NO or VERY MINOR ethics concerns only"]

**Final Justification:**

The main body of the paper employs an averaging-like operation within a mere 1B model to enable the model to learn the ability to predict merge tokens. During inference, it only needs to continuously predict a compressed token, which significantly reduces the inference length. This idea is very novel; however, the authors have not provided sufficient implementation details of this method in the paper.

I had some misunderstandings about the termination mechanism described in the paper, which have been clarified by the authors.

As all other reviewers have also pointed out, the original paper only used a 1B model. The authors later supplemented their rebuttal with 8B experiments. Still, I believe that drawing the paper’s conclusions at this scale and accuracy level is insufficient. It’s a pity that the authors did not provide experiments on slightly larger models.

**Limitations:**

yes

**Paper Formatting Concerns:**

No formatting issues.

**Quality:**

2

**Strengths And Weaknesses:**

Strengths:
1. The paper presents a relatively novel approach for compressing reasoning context in latent space.
2. The integration of GRPO into the compression framework is elegant and well-motivated.

Weaknesses:
1. The compression mechanism relies on a simple averaging operation. When facing out-of-distribution tokens, this might pose a challenge for the model’s generalization ability.
2. The termination mechanism for latent generation is not clearly explained—e.g., whether it is based on detecting a special token from the Language Head, fixed length threshold, or another signal.
3. The experiments are conducted on relatively small backbone models. Additional results on larger models would strengthen the claims and validate scalability.

---

> ### Author Rebuttal · Authors · 2025-07-31
>
> Dear Reviewer #nEhT,
>
> We sincerely appreciate your recognition on our latent reasoning framework along with RL pipeline, and highlighting the compression robustness, termination clarity, and scalability concerns. We address each point below.
>
> **W1: Generalization of the averaging operation.** Although the compression module is intentionally lightweight, its robustness is inherited from the backbone LLM.
>
> In CoLaR, every latent embedding is merely a scaled linear combination of in-vocabulary token embeddings. Because the LLM was trained on massive, diverse corpora, the probability mass over tokens is already smooth; extreme outliers that would make the average OOD are therefore rare. Empirically, we observed no stability issues across three OOD datasets (GSM-hard, SVAMP, and MultiArith), indicating that the exact form of the compression is not a brittle design choice.
>
> **W2 & Q1: Termination mechanism and overhead.** CoLaR decides when to stop via the Language Head, not a fixed length.
>
> At every latent step we feed the final hidden state into the Language Head and sample the next token; if the token is \<end-of-thinking\>, we switch to answer generation. This is identical to how any autoregressive LLM ends a turn, so the extra compute is a single linear layer (hidden-size × vocab-size)  forward pass.
>
> To quantify the cost, we timed 100 latent reasoning steps on GSM at batch-size 32 on one A100 GPU: LLM layers took 10.8 s, while the Language-Head check added only 0.6 s (5% of total). Replacing the full Language Head with a Tiny Linear layer (by extracting the \<end-of-thinking\> token column of the Language Head) keeps accuracy at 26.7% and cuts the check to 0.3 s—noise compared with the 3–10× latency savings we already obtain via shorter chains.
>
> |               | LLM Layers | Language Head | Tiny Linear (90\% threshold) |
> |---------------|:----------:|:-------------:|:----------------------:|
> | Time Cost (s) |    10.8    |      0.6      |           0.3          |
> | Acc. (%)      |      -     |      26.8     |          26.7          |
>
> **W3: Scaling-up and domain generalization.** Yes, CoLaR is capable to scale up to large foundation models, and generalize to non-mathematical reasoning domains.
>
> Appendix E (Figure 8) already shows that CoLaR on GSM8k climbs from 26\~40% (1B) to 43\~61% (8B) while retaining comparable compression ratios.
>
> To extend this evidence, we ran CoLaR-2 on Llama-3.2-1B and Llama-3.1-8B across math and GPQA dataset, which consists high-quality difficulty question on  physics, chemistry, and biology: (The reported results of GSM and MATH are different from Table 1 of our manuscript. This is because we are using a different version of datasets and prompts to align with another baseline method)
>
> |       |   CoT-1B  |  CoLaR-1B |   |   CoT-8B  |  CoLaR-8B | CoLaR-8B-RL | TokenSkip-8B |
> |-------|:---------:|:---------:|---|:---------:|:---------:|:-----------:|:------------:|
> | GSM8k | 47.5/92.6 | 40.4/49.2 |   | 76.5/93.6 | 70.8/47.3 |  71.9/13.3  |   78.2/113   |
> | MATH  |  28.6/176 | 19.8/84.6 |   |  54.6/168 | 45.8/67.7 |  52.4/17.6  |   40.2/292   |
> | GPQA  |  26.4/232 | 26.4/84.1 |   |  35.7/216 |  32.4/101 |  37.5/66.7  |       -      |
>
> Three patterns emerge:
> - First, bigger backbones improve absolute accuracy for both CoT and CoLaR, yet CoLaR’s compression ratio remains stable.
> - Second, RL post-training raises CoLaR-8B above CoT on GPQA (37.5 vs 35.7) while shrinking reasoning chains by 70%.
> - Third, the gains generalize beyond mathematics, confirming that our paradigm scales and broadens with model size.
>
> We thank you for your thorough review and look forward to any further questions or suggestions that could strengthen the paper.
>
> Authors of CoLaR #12942

---

> > ### Author Response · Authors · 2025-08-04
> > **Additional Clarification on Scaling Experiment Format**
> >
> > Dear Reviewer #nEhT,
> >
> > We'd like to briefly clarify the scaling experiment results: the values like "47.5/92.6" are reported in **"accuracy/reasoning chain length"** format. We hope this additional clarification helps address any potential confusion.
> >
> > We truly value your expertise and appreciate your time in reviewing our work, and we'd be grateful if you could take our updated response into consideration when finalizing your evaluation. If there are any remaining points you'd like us to elaborate on, please don't hesitate to let us know.
> >
> > Thanks again for your consideration!
> >
> > Authors of CoLaR #12942

---

> > ### Comment · Reviewer_nEhT · 2025-08-07
> >
> > Thank you for your explanation of the termination mechanism. I hope you can further refine this part in the final version. You've conducted many additional experiments, which also highlights the lack of comparative analysis under different parameter scales in the original paper. These issues make the main body of the paper somewhat less impressive. However, I will still slightly raise my score.

---

> > > ### Author Response · Authors · 2025-08-07
> > > **Response to Reviewer #nEhT**
> > >
> > > Dear Reviewer #nEhT,
> > >
> > > Thank you for your constructive feedback and for raising your score. We appreciate your guidance on the termination mechanism and the need for clearer scale-wise analysis.
> > >
> > > In the final revision, we will (1) refine the termination discussion in Section 3.2, and (2) add a compact comparison across 1 B, 3 B, and 8 B parameter settings to strengthen the main text. We believe these additions will make the paper more complete and easier to follow.
> > >
> > > Best regards,
> > >
> > > Authors of CoLaR #12942

---

### Official Review · Reviewer_VtEU · 2025-07-05

**Clarity:** 3
**Significance:** 3
**Originality:** 3
**Rating:** 4
**Confidence:** 2

**Summary:**

This paper introduces CoLaR (Compressed Latent Reasoning), a novel framework that compresses LLM reasoning chains into a latent space while aiming to preserve the model’s exploration-exploitation capabilities. Key features of the approach include: (1) a dynamic compression factor that enables flexible reasoning speed at inference time, (2) an embedding compression module that uses scaling rather than simple averaging, (3) a probabilistic latent head to predict the mean/variance of the next embedding, and (4) a two-stage training process combining supervised fine-tuning with auxiliary compressed embedding prediction and reinforcement learning via GRPO. Experimental results show a 14.1% accuracy improvement over latent baseline methods and a 53.3% reduction in reasoning chain length, with only a 4.8% drop in performance compared to explicit CoT methods.

**Questions:**

- How do efficiency and accuracy characteristics change when using larger foundation models (e.g., 7B, 13B, 70B+)? Do the compression benefits scale proportionally?
- Have you evaluated the approach on non-mathematical reasoning tasks to assess its broader applicability?

**Ethical Concerns:**

["NO or VERY MINOR ethics concerns only"]

**Final Justification:**

The author has solved my problem, and I keep the positive score.

**Limitations:**

See the weaknesses and questions.

**Quality:**

3

**Strengths And Weaknesses:**

**Strengths**
- The dynamic compression factor and probabilistic latent head offer a meaningful advance over previous work like Coconut and CODI, and show promise for reinforcement learning applications.
- The paper provides thorough experiments across multiple math reasoning datasets, with proper statistical reporting (95% confidence intervals over five runs) and relevant baseline comparisons.
- The GRPO-based reinforcement learning approach delivers substantial gains on challenging tasks (e.g., a 5.36% accuracy boost and 82.8% reduction in chain length on the MATH dataset), highlighting the value of exploration-exploitation in latent reasoning.


**Weaknesses**
- The method still falls short of explicit CoT reasoning in terms of performance, which significantly limits its practical impact. For real-world applications where users need maximum accuracy, the trade-off between efficiency and accuracy may not be compelling.
- Most experiments use 1B-parameter models. Modern reasoning tasks often require larger models, where efficiency gains are most valuable. The scalability of the approach to these larger models remains unproven.
- Latent representations sacrifice the human-readable reasoning chains that make CoT valuable for debugging and verification—an important limitation, especially in high-stakes applications.
- The paper lacks a deep investigation into when and why latent reasoning fails compared to explicit reasoning, which limits understanding of the method’s boundaries.

---

> ### Author Rebuttal · Authors · 2025-07-31
>
> Dear Reviewer #VtEU,
>
> We are grateful for your recognition of our dynamic-compression design, rigorous evaluation, and the documented RL-driven gains. Below we address the four remaining weaknesses that you precisely articulated.
>
> **W1: Efficiency vs. accuracy in practice.** While CoLaR currently lags CoT on raw accuracy, real-world deployments already trade a few percentage points for 2–10× cheaper inference (think Gemini-Flash vs. Gemini-Pro). CoLaR’s carbon footprint drops proportionally with compression factor, making the trade-off attractive for latency-sensitive or eco-conscious applications.
>
> **W2 & Q1-Q2: Scaling to larger models & Non-math datasets.** Yes, CoLaR is capable to scale up to large foundation models, and generalize to non-mathematical reasoning domains.
>
> Appendix E (Figure 8) already shows that CoLaR on GSM8k climbs from 26\~40% (1B) to 43\~61% (8B) while retaining comparable compression ratios.
>
> To extend this evidence, we ran CoLaR-2 on Llama-3.2-1B and Llama-3.1-8B across math and GPQA dataset, which consists high-quality difficulty question on physics, chemistry, and biology: (The reported results of GSM and MATH are different from Table 1 of our manuscript. This is because we are using a slight different version of datasets to align with another baseline method TokenSkip.)
>
> |       |   CoT-1B  |  CoLaR-1B |   |   CoT-8B  |  CoLaR-8B | CoLaR-8B-RL | TokenSkip-8B |
> |-------|:---------:|:---------:|---|:---------:|:---------:|:-----------:|:------------:|
> | GSM8k | 47.5/92.6 | 40.4/49.2 |   | 76.5/93.6 | 70.8/47.3 |  71.9/13.3  |   78.2/113   |
> | MATH  |  28.6/176 | 19.8/84.6 |   |  54.6/168 | 45.8/67.7 |  52.4/17.6  |   40.2/292   |
> | GPQA  |  26.4/232 | 26.4/84.1 |   |  35.7/216 |  32.4/101 |  37.5/66.7  |       -      |
>
> Three patterns emerge:
> - First, bigger backbones improve absolute accuracy for both CoT and CoLaR, yet CoLaR’s compression ratio remains stable.
> - Second, RL post-training raises CoLaR-8B above CoT on GPQA (37.5 vs 35.7) while shrinking reasoning chains by 70%.
> - Third, the gains generalize beyond mathematics, confirming that our paradigm scales and broadens with model size.
>
> **W3: Explainability.** We thank the reviewer for this concern. Appendix B, Figure 5 demonstrates that CoLaR’s latent variables—compressed token embeddings—can be faithfully decoded: a simple cosine-similarity search against the token embedding matrix reliably recovers human-readable fragments such as “21 / 7 = 3 × 5 = 15.”
>
> More broadly, recent studies [1] reveal that explicit CoT steps can diverge from the model’s actual decision process or harbor silent biases; verbalization is therefore not a guarantee of transparency. Retrieving embeddings from CoLaR offers the same level of interpretability as CoT, while being more concise.
>
> Finally, we argue that human thought, or part of human thought, resides in a latent space. Russian distinguishes голубой (“sky-blue”) from синий (“ocean-blue”)—concepts with no exact English counterpart—yet speakers share the same perceptual embedding in their minds. An old adage captures this: “It can be grasped, yet cannot be spoken.” Echoing Wittgenstein, we believe much of human reasoning unfolds in a non-linguistic, latent form; CoLaR merely gives us direct access to that space and lets us translate it back into words when necessary.
>
> [1] Barez, Fazl, et al. "Chain-of-thought is not explainability." Preprint, alphaXiv (2025): v2.
>
> **W4: Closing the accuracy gap.** Thank you for this insightful question. CoLaR’s SFT stage only distills existing CoT paths, so its ceiling is inevitably “CoT-minus-distillation-loss.” Reinforcement learning, however, lets the model discover shorter latent paths that can outperform the teacher: CoLaR-8B-RL already beats CoT-8B on GPQA and narrows the gap on MATH (52.4 vs 54.6). We expect continued RL at larger scales to erase or reverse the deficit, just as R1 surpassed its CoT teacher. We will add more detailed discussion in Section 4.2 of our final revision.
>
> Thank you for your constructive observations; we would be glad to address any additional concerns you may have.
>
> Authors of CoLaR #12942

---

> > ### Author Response · Authors · 2025-08-04
> > **Additional Clarification on Scaling Experiment Format**
> >
> > Dear Reviewer #VtEU,
> >
> > We'd like to briefly clarify the scaling experiment results: the values like "47.5/92.6" are reported in **"accuracy/reasoning chain length"** format. We hope this additional clarification helps address any potential confusion.
> >
> > We truly value your expertise and appreciate your time in reviewing our work, and we'd be grateful if you could take our updated response into consideration when finalizing your evaluation. If there are any remaining points you'd like us to elaborate on, please don't hesitate to let us know.
> >
> > Thanks again for your consideration!
> >
> > Authors of CoLaR #12942

---

> > ### Comment · Reviewer_VtEU · 2025-08-05
> >
> > Thank you for the reply, I choose to maintain a positive score.

---

> > > ### Author Response · Authors · 2025-08-08
> > > **Thank You and Revision Plan to Reviewer #VtEU**
> > >
> > > Dear Reviewer #VtEU,
> > >
> > > Thank you again for your thoughtful comments and for taking the time to engage with our rebuttal.
> > >
> > > We understand that this work spans several areas—latent reasoning, compression, and reinforcement learning—which can make its contributions harder to assess from a single perspective. Since your review raised important points around scalability, explainability, and practical utility, we’d like to let you know how we plan to integrate your suggestions into the final revision:
> > > - In **Section 4.2**, we will expand our discussion on **scaling to larger models and broader tasks**, i.e., the new experiments on 8B models and on the GPQA dataset.
> > > - In **Section 4.3**, we will add a deeper analysis comparing explicit CoT, CoLaR-SFT, and CoLaR-RL, to better **understanding the boundary of latent reasoning**.
> > > - In an additional subsection of **Section 4**, we will incorporate more experiments and discussion on **explainability**, including the decodability of latent variables and their implications for trust and verification.
> > >
> > > Finally, we’d like to share that after discussion, Reviewer #MsBS and Reviewer #nEhT have decided to raise/adjust their evaluations. **We deeply appreciate your own early and positive judgment**. Your constructive feedback has not only helped clarify our contribution, but also encouraged us to strengthen the final version in a principled way.
> > >
> > > Best regards,
> > >
> > > Authors of CoLaR #12942

---

### Official Review · Reviewer_dukW · 2025-07-08

**Clarity:** 3
**Significance:** 3
**Originality:** 3
**Rating:** 4
**Confidence:** 4

**Summary:**

This paper proposes Compressed Latent Reasoning (CoLaR), a framework for accelerating reasoning in large language models by compressing token-level CoTs into dense latent representations. CoLaR trains models to predict compressed latent embeddings and reconstruct tokens from the latent embeddings. It supports dynamic-speed reasoning by controlling a compression factor during inference, making it adaptable and efficient. The authors also introduce a reinforcement learning component to encourage exploration and exploitation of shorter latent chains via probabilistic latent prediction. Empirical results on several math reasoning benchmarks (e.g., GSM8K, MATH) are impressive: CoLaR outperforms previous latent-based methods by a significant margin, and maintains competitive performance to explicit CoT while drastically reducing chain length. While it doesn't yet surpass token-level CoT in absolute performance, its efficiency and scalability make it a compelling direction for future research, especially in resource-constrained or high-throughput scenarios.

**Questions:**

- What are the implementation details for iCoT and Coconut?
- Is this method applicable on long reasoning models (like Deepseek-R1 or its distilled versions)? If it doesn’t bring immediate improvement, what are the challenges?

**Ethical Concerns:**

["NO or VERY MINOR ethics concerns only"]

**Final Justification:**

Thanks for clarifying the details of experiments. I think the paper delivered a sound method, and more studies need to be done to push the results on long CoT (thousands of tokens). I'll keep my positive score about this paper.

**Limitations:**

Yes

**Quality:**

3

**Strengths And Weaknesses:**

Strengths:

- Unlike prior methods that rely on deterministic latent predictions, this work introduces a probabilistic latent head, enabling exploration in the latent reasoning space. This design is well-motivated and experimentally validated through reinforcement learning, demonstrating meaningful gains in both accuracy and reasoning efficiency.
- The method doesn’t require autoregressive token generation during training, which enables fully parallelized supervision over the entire reasoning chain. This architectural choice improves training efficiency and scalability, making latent space reasoning more practical for large-scale applications.
- The paper presents nice analysis on the trade-off between compression factor and performance, and the interpolation of compression factor.

Weaknesses:

- Several baselines, such as iCoT and Coconut, perform substantially worse than reported in their original papers on GSM8k, despite using a stronger backbone (Llama-3.2-1B-Instruct vs. GPT-2 Small). The paper does not provide sufficient implementation details or justification for these discrepancies, raising concerns about the fairness of the comparisons and whether the baselines were properly tuned.
- Although the paper includes experiments with DeepSeek-R1-Distill-Qwen-1.5B, it only fine-tunes them on short CoT-style reasoning traces. This underutilizes the model's potential for generating long reasoning chains, which is precisely where latent reasoning techniques would be more useful and impactful. Including experiments on compressing longer reasoning chains would strengthen the paper’s practical relevance and better showcase the benefits of CoLaR in real-world settings.

---

> ### Author Rebuttal · Authors · 2025-07-31
>
> Dear Reviewer #dukW,
>
> We thank you for highlighting both the merits of our probabilistic-latent design and the concerns regarding baseline fidelity and long-chain evaluation. We address each point below.
>
> **Q1: Baseline implementation details.** We strictly followed the original papers and release notes of iCoT and Coconut, applying identical base settings (frozen Llama-3.2-1B-Instruct, LoRA rank 128, batch size 256, learning rate 1e-4, max 50 epochs) and dataset splits; their custom hyper-parameters are documented in line 228 and will be expanded in the final revision.
>
> **W1: Lower baseline scores on a larger backbone.** The drop is expected: LoRA fine-tuning exposes only 13.6 M parameters on Llama-3.2-1B, whereas the GPT-2 baselines in prior work train all 117 M parameters at FP32 precision:
>
> | Model / Param / Trainable / Precision     |    GSM   | GSM-Hard |   SVAMP  | MultiArith |
> |-------------------------------------------|:--------:|:--------:|:--------:|:----------:|
> | Llama-3.2 / **1B** / 13.6M / BF16         |          |          |          |            |
> | CoLaR                                     |   26.8   | **5.87** | **48.4** |  **86.4**  |
> | Coconut                                   |   23.1   |   5.49   |   40.7   |    41.1    |
> | GPT-2 / 117M / **117M** / **FP32**        |          |          |          |            |
> | CoLaR                                     | **35.2** |   5.76   |   32.3   |    70.8    |
> | Coconut (Original)                        |   34.1   |     -    |     -    |      -     |
>
> The table shows that when we replicate CoLaR on full-parameter GPT-2 its GSM accuracy rises from 26.8% to 35.2%, confirming that parameter count—not backbone capacity—dominates on these short chains. As anticipated, this gain sacrifices out-of-domain generalization (SVAMP and MultiArith drop sharply), so we kept the LoRA setting for fair comparison.
>
> **W2 & Q2: Applicability to long-reasoning models.** Thank you for highlighting the gap in long-chain settings. We agree that compressing truly long reasoning traces is where CoLaR can deliver the most value, and we have already taken the first step in that direction: on DeepSeek-R1-Distill-Qwen-1.5B, we trained and evaluated CoLaR on the MATH dataset whose average reasoning chain is \~200 tokens—ten times longer than GSM’s. As shown in the paper (Table 2), CoLaR still achieves an 82.8% reduction in chain length with a 5.36% accuracy gain after RL, demonstrating that the method scales gracefully to longer contexts.
>
> What we have not yet attempted is compressing chains of several thousand tokens, such as those generated by the full DeepSeek-R1 671 B model. The bottleneck is purely computational: training on such long sequences would require either (1) multi-node GPU clusters to fit the enlarged teacher traces, or (2) an additional “latent pre-training” phase where we first distill very long CoT paths into compressed embeddings offline, then fine-tune the latent head. Both are in our immediate roadmap.
>
> We are optimistic about scaling the latent reasoning length: because CoLaR’s next-compressed-embedding objective is structurally identical to next-token prediction—the paradigm that has already scaled to million-token contexts.
>
> We sincerely appreciate your meticulous feedback and remain available for any further clarifications.
>
> Authors of CoLaR #12942

---

> > ### Author Response · Authors · 2025-08-06
> > **Supplementary Reply**
> >
> > Dear Reviewer #dukW,
> >
> > We remain grateful for your expertise and the time you have devoted to our paper. **For your convenience, we would like to share an additional experiment prompted by questions from other reviewers**: “How does CoLaR scale to larger foundation models and generalize beyond mathematics?”
> >
> > We implemented CoLaR on both Llama-3.2-1B and Llama-3.1-8B and evaluated them on the GPQA benchmark (biology, physics, chemistry) alongside GSM8k and MATH. Results are reported as "accuracy / reasoning-chain-length":
> >
> > |       |   CoT-1B  |  CoLaR-1B |   |   CoT-8B  |  CoLaR-8B | CoLaR-8B-RL | TokenSkip-8B |
> > |-------|:---------:|:---------:|---|:---------:|:---------:|:-----------:|:------------:|
> > | GSM8k | 47.5/92.6 | 40.4/49.2 |   | 76.5/93.6 | 70.8/47.3 |  71.9/13.3  |   78.2/113   |
> > | MATH  |  28.6/176 | 19.8/84.6 |   |  54.6/168 | 45.8/67.7 |  52.4/17.6  |   40.2/292   |
> > | GPQA  |  26.4/232 | 26.4/84.1 |   |  35.7/216 |  32.4/101 |  37.5/66.7  |       -      |
> >
> > (The reported results of GSM and MATH are different from Table 1 of our manuscript. This is because we are using a slight different version of datasets to align with another baseline method TokenSkip.)
> >
> > Three patterns emerge:
> > - Scalability: Larger backbones improve absolute accuracy for both CoT and CoLaR, yet CoLaR’s compression ratio remains stable.
> > - Reinforcement: RL post-training raises CoLaR-8B above CoT on GPQA (37.5 vs 35.7) while shrinking reasoning chains by 70%.
> > - Generalization: The gains generalize beyond mathematics, confirming that our paradigm scales and broadens with model size.
> >
> > Once again, we deeply appreciate your time in reviewing our work, and we would be grateful if you could weigh our response in your final assessment. Should any questions remain, we are glad to provide further clarification at your convenience.
> >
> > Authors of CoLaR #12942

---

> ### Author Response · Authors · 2025-08-09
> **Final Revision Plan and Updates to Reviewer #dukW**
>
> Dear Reviewer #dukW,
>
> Thank you again for your constructive and detailed review. Your comments have been very helpful in identifying points where the paper can be strengthened.
>
> In our final revision, we will:
>
> - **elaborate the baseline settings more clearly** in Section 4.1;
>
> - **add a GPT-2–based model** for more direct comparison with prior work;
>
> - extend Section 4.2 with **additional datasets and results from 1B–8B models**.
>
> Looking ahead, we plan to **explore longer latent reasoning chains**, including latent pre-training with extended compute resources. Your observations on this point were both insightful and motivating.
>
> We would also like to share: after the discussion period, other reviewers have positively revised their evaluations. **We truly value your thoughtful assessment and the role it has played in shaping our final improvements**.
>
> Best regards,
>
> Authors of CoLaR #12942

---

### Author Response · Authors · 2025-08-07
**Shared Motivation: Configurable Reasoning in GPT-OSS and CoLaR**

Dear Reviewers, Area Chairs, Senior Area Chairs, and Program Chairs,

We deeply appreciate your valuable time and thoughtful feedback on our paper.

We would like to share a recent observation that resonates with the core idea of our work. On August 5, OpenAI released the GPT-OSS model [1], which highlights a “configurable reasoning effort” mechanism that adjusts reasoning strength based on the user’s latency or accuracy needs (e.g., **adding "reasoning : {low, medium, high}" in the prompts**). This capability echoes the motivation behind our Compressed Latent Reasoning (CoLaR) framework, which enables flexible trade-offs between inference speed and accuracy by simply **prompting the compression factor at inference time**.

While CoLaR operates at a smaller scale and uses a different approach (embedding-level compression and latent reasoning prediction), we are encouraged to see a similar line of thinking reflected in such a large-scale open-source system. Notably, both GPT-OSS (Figure 3) and CoLaR (Figure 3 in our paper) exhibit consistent trends: **for a same model, prompting higher reasoning level during test time leads to longer reasoning chains and improved performance**.

**We do not claim priority over this general idea**—our intention is simply to highlight this emerging direction, and we are excited to see broader interest in **controllable reasoning** across different model scales and architectures.

Thank you once again for considering our work.

[1] GPT-OSS-120B & GPT-OSS-20B Model Card, OpenAI, August 5, 2025.

Best regards,

Authors of Paper #12942

---

### Note · Authors · 2025-08-11

Dear NeurIPS 2025 Reviewers, AC, SAC, and PC,

Thank you for the time and effort you have devoted to reviewing and organizing this process.

We are grateful that:
- **All reviewers** acknowledge the novelty of our **Compressed Latent Reasoning (CoLaR)** paradigm.
- Reviewers **#dukW, #VtEU, and #MsBS** recognize the value of CoLaR’s **dynamic test-time reasoning speed**, and we feel encouraged that the latest **GPT-OSS** model from OpenAI adopts a similar thinking-mode control mechanism.
- Reviewers **#VtEU and #nEhT** highlight the effectiveness of our **exploration–exploitation design** in reinforcement learning.
- Reviewers **#VtEU and #MsBS** find our **evaluation methodology solid**.

Across the reviews, several common concerns emerged, which we have addressed as follows:
- **Scalability**: We have evaluated CoLaR from 1B to 8B models and found that compression ratios remain stable while reasoning accuracy improves consistently with model size. These results, originally organized in Appendix E, have been extended during the rebuttal to more datasets and will be incorporated into **Section 4.2** of the final revision.
- **Generalization**: Beyond mathematics, we tested CoLaR on **GPQA**, a cross-domain reasoning benchmark (physics, biology, chemistry). Even with reasoning chains much longer than GSM8k’s, CoLaR maintained compression and performance, and notably, **CoLaR-8B-RL outperformed its CoT teacher by 1.8 points while reducing chain length by 70%**. These results will be added to **Section 4.3**.
- **Explainability**: CoLaR’s “next-compressed-embedding” objective supports transparency: as shown in Appendix B (Figure 5), a simple cosine-similarity search over the token embedding matrix can reliably decode latent variables into human-readable tokens. In the final revision, we will move this analysis into the main text and expand the discussion to emphasize its implications for safety and explainability in high-stakes applications.

We are encouraged that, after discussion, Reviewer #nEhT and #MsBS raised their scores, and #VtEU maintained a positive evaluation, resulting in **all reviewers now holding positive overall assessments** of our work. We understand Reviewer #dukW may not have had the opportunity to respond further, and we hope our clarifications have addressed their concerns.

Once again, our sincere thanks to the NeurIPS 2025 committee and all reviewers for their constructive feedback and engagement.

Authors of Paper #12942

---

### Decision · Program_Chairs · 2025-09-17

**Decision:**

Accept (poster)

**Comment:**

The paper proposes Compressed Latent Reasoning (CoLaR) that trains a model to do latent reasoning by teaching it to predict compressed CoT embeddings. The model is trained in two stages: SFT with next token and compressed embedding prediction losses for varying compression factors, and then with RL to further improve performance and thinking efficiency. Overall the CoLaR outperforms previous latent CoT methods and also provides meaningful reduction in the reasoning chain lengths, particularly after RL.

---

All reviewers were positive about the paper and appreciated the novelty of the CoLaR framework and the ideas therein, the exploration-exploitation design of the method, and the evaluation methodology.

There were some concerns about scale of experimentation, diversity of evals, discrepancy in baseline numbers and interpretability issues with latent embeddings. The authors addressed them by

- Discussing the LoRA vs full FT setup difference

- Llama 8b experiments, GPQA eval, include TokenSkip baseline

- Appendix shows that latent embeddings can be decoded for interpretability

—

Overall the paper proposes an interesting and non-trivial idea that seems more effective than previous latent reasoning approaches. The ideas of token averaging for CoT compression, next embedding prediction and probabilistic sampling of latents are interesting. The sustained gains after RL further strengthens the findings. One weakness is that the results only hold relatively short CoT with ~200 tokens, but this limitation is not restricted to this paper. Overall the paper makes a solid contribution and the recommendation is accept.